# Strategies for Radioiodine Treatment: What’s New

**DOI:** 10.3390/cancers14153800

**Published:** 2022-08-04

**Authors:** Clotilde Sparano, Sophie Moog, Julien Hadoux, Corinne Dupuy, Abir Al Ghuzlan, Ingrid Breuskin, Joanne Guerlain, Dana Hartl, Eric Baudin, Livia Lamartina

**Affiliations:** 1Endocrinology Unit, Department of Experimental and Clinical Biomedical Sciences “Mario Serio”, University of Florence, 50139 Florence, Italy; 2Service d’oncologie Endocrinienne, Département d’Imagerie Médicale, Gustave Roussy, 112 rue Edouard Vaillant, 94805 Villejuif, France; 3UMR 9019 CNRS, Université Paris-Saclay, Gustave Roussy, 94800 Villejuif, France; 4Département de Biologie et Pathologie Médicales, Gustave Roussy, 112 rue Edouard Vaillant, 94805 Villejuif, France; 5Département Anesthésie Chirurgie et Interventionnel, Gustave Roussy, 112 rue Edouard Vaillant, 94805 Villejuif, France

**Keywords:** radioiodine, thyroid cancer, quality of life, risk assessment, overtreatment, redifferentiation

## Abstract

**Simple Summary:**

Radioactive iodine treatment is the oldest targeted therapy for differentiated thyroid cancer. It can be used for normal thyroid remnant ablation (in order to increase the sensitivity and the specificity of the serum marker thyroglobulin), as an adjuvant treatment (in order to improve recurrence-free survival), or to treat radioiodine avid residual disease. Thanks to the use of sensible diagnostic tools, reliable prognostic classifications, and molecular profiling, the indication and modalities of radioiodine treatment are shifting from a one-size-fits-all to a tailored approach. This review provides insights into the most recent and high-quality evidence relating to radioactive iodine treatment.

**Abstract:**

Radioiodine treatment (RAI) represents the most widespread and effective therapy for differentiated thyroid cancer (DTC). RAI goals encompass ablative (destruction of thyroid remnants, to enhance thyroglobulin predictive value), adjuvant (destruction of microscopic disease to reduce recurrences), and therapeutic (in case of macroscopic iodine avid lesions) purposes, but its use has evolved over time. Randomized trial results have enabled the refinement of RAI indications, moving from a standardized practice to a tailored approach. In most cases, low-risk patients may safely avoid RAI, but where necessary, a simplified protocol, based on lower iodine activities and human recombinant TSH preparation, proved to be just as effective, reducing overtreatment or useless impairment of quality of life. In pediatric DTC, RAI treatments may allow tumor healing even at the advanced stages. Finally, new challenges have arisen with the advancement in redifferentiation protocols, through which RAI still represents a leading therapy, even in former iodine refractory cases. RAI therapy is usually well-tolerated at low activities rates, but some concerns exist concerning higher cumulative doses and long-term outcomes. Despite these achievements, several issues still need to be addressed in terms of RAI indications and protocols, heading toward the RAI strategy of the future.

## 1. Differentiated Thyroid Cancer: Mortality and Recurrence Risks

Differentiated thyroid cancer (DTC) represents the most common endocrine tumor, showing an increasing incidence trend during the last decades [1]. Papillary thyroid carcinoma (PTC) is the most common histological subtype [2] and a large number of new cases are related to small and indolent PTC, often found incidentally due to the increased use of diagnostic investigations [3,4]. Despite the steep increase in DTC incidence, the specific five-year-survival rates are about 98% [1] and the related age-adjusted mortality remains stable (about 0.5 per 100,000 per year), with no further expected change according to recent global estimates [4]. The extremely good prognosis of DTC and an increased awareness of the risks of treatments possibly resulting in life-long quality of life (QoL) impairment, has promoted a wide revision of diagnostic and management strategies [2]. This management shift is reflected in the recently observed decrease in indolent tumors (PTC ≤ 1.0 cm and localized disease) incidence in the United States (14.5 vs. 13.48 per 100,000 person-years in 2014 and 2018, respectively) [5] with a stable rate of loco-regional and advanced DTC [5]. In Europe, a similar trend emerged from the French registries, where a decrease in microcarcinomas and a slight, but constant, increase in larger DTC (>40 mm) diagnoses were observed [6].

The upfront therapeutic approach for DTC classically relies on surgery followed by radioiodine (RAI) treatment. The recourse to RAI has deeply changed throughout DTC history, evolving from a “one-size-fits-all” approach to a more selective choice. In fact, the observation of RAI effectiveness in reducing DTC recurrences and mortality initially promoted its use for all cases, irrespective of the disease extension and recurrence risks [7]. At that time, it was considered impossible to carry out adequately powered randomized trials in lower-risk thyroid cancer due to their indolent nature and the relative rarity of the disease. Two European academic studies, the ESTIMABL [8] and the HiLo [9] trial broke this paradigm, paving the way for an evidence-based approach for lower-risk DTCs.

Moreover, observational data on large cohorts found no survival benefit for lower stage patients (corresponding to patients at lower risk) treated with or without RAI [10], and several retrospective works highlighted the good prognosis of small and localized DTC, treated or not treated with RAI, questioning the paradigm of indiscriminate RAI use [11].

Accurate disease risk stratification can assist in the selection of the patients who may benefit from RAI therapy. Two key factors need to be addressed in patients’ initial prognostic assessment: the risk of death, provided by the TNM stage from the American Joint Committee on Cancer (AJCC) [12], and the risk of disease recurrence, based on the American Thyroid Association (ATA) [2] guidelines. These two classifications complement each other since the AJCC staging only predicts long-term mortality—usually very low for DTC—and is not designed to predict tumor recurrence; while ATA classification provides a valid medium-to-long term stratification for structural recurrent disease [13,14] (Table 1). It is also worth noting that patients with lower-stage DTC (i.e., at low risk of death) can be at high risk of recurrence [13].

The ATA risk of recurrence classification includes three categories: (i) low-risk DTC (risk of recurrence being less than 5%); (ii) intermediate-risk (less than or equal to 20%); (iii) high-risk DTC (more than 20%) [2]. Within each class, we should also consider some additional features (e.g., the number or the extent of lymph node involvement, or the histological variant), which provide auxiliary prognostic information. There is quite a uniform consensus on RAI use for two opposing categories of DTC: on one hand, this therapy is considered avoidable for (very) low-risk DTC, i.e., intrathyroidal papillary microcarcinoma (PTMC), with an expected recurrence rate of less than 2% [14,15]; on the other hand, the use of RAI is mandatory for high risk-patients (Table 1) [2], who represent the rare cases burdened by a higher disease-mortality rate. Between these two extremes, we find the largest group of low and intermediate cases (about 70% of all DTC), where the use of radioiodine is still controversial.

## 2. Radioiodine Treatment Goals

The administration of RAI pursues different objectives. According to the 2015 ATA guidelines [2], the goals of RAI are: (i) ablative, i.e., the destruction of normal thyroidal remnants, in order to simplify follow-up management; (ii) adjuvant, i.e., the destruction of microscopic tumor foci, in order to improve disease-free survival; (iii) therapeutic, where pathological tissue is expected or diagnosed; (iv) to provide a whole-body evaluation by whole-body scan/Single Photon Emission Computed Tomography (WBS/SPECT-CT), which can potentially change the initial staging, uncovering loco-regional or distant tumor foci. The 2022 European Consensus statement [16] was signed by four societies involved in thyroid cancer management: i.e., the American Thyroid Association (ATA), the European Thyroid Association (ETA), the European Association of Nuclear Medicine, and the Society of Nuclear Medicine and Molecular Imaging. Their purpose was to clarify some key points in RAI treatment indications for DTC and a revision of the traditional nomenclature was also proposed. In particular, they suggested replacing the conventional term “ablative RAI”—i.e., the first radioiodine administration—with the general label of “radioiodine therapy”, which includes all of the potential therapeutic goals: ablative, adjuvant, and (consequently) therapeutic [16].

In order to enhance the administration of radioiodine, a significant increase in thyrotropin (TSH) levels is needed to improve cellular uptake. The required and most accepted value of TSH is conventionally set above 30 mU/L [2]. The ways to obtain this value are: by thyroid hormone withdrawal (THW) performed for four weeks, or after a period of liothyronine (LT3), to reduce deep hypothyroidism sides effects (about for 2–3 weeks before a complete THW); or by recombinant human TSH (rhTSH) stimulation. Before rhTSH availability, THW was the standard protocol for RAI preparation. Hypothyroidism subsequent to THW is associated with transient impairment of QoL and cognitive status, general discomfort, and several adverse events, including cardiovascular and renal function deterioration [2,17,18,19,20]. Thereafter, the equivalence of rhTSH and THW for ablation preparation has been suggested by several studies [20,21,22,23], also showing the advantages of rhTSH in terms of preserving QoL [17,20,21,22,23], lower absorbed RAI activity for abdominal organs [24], and the significant reduction of global radiation exposure in sensitive organs, including bone marrow [25].

### 2.1. Remnant Ablation

As explained above, the most widespread application of radioiodine is the ablation of residual thyroid tissue, also known as radioiodine remnant ablation (RRA), and its therapeutic protocol underwent numerous revisions over time [16,26].

Serum thyroglobulin (Tg) is a tissue-specific rather than a tumor-specific marker, hence the elimination of all normal thyroid remnants renders the interpretation of this serum marker easier. The main purpose of RRA is to simplify follow-up management, enhancing the diagnostic performance of Tg concentration, rather than improving recurrences or survival [27,28]. Another possible goal of RRA is to facilitate the neutralization of anti-Tg antibodies (TgAb), which could produce interferences in Tg measurement resulting in false-negative, or more rarely, in false-positive results rendering the Tg value not interpretable. Two randomized clinical trials (RCT), ESTIMABL1 [8] and HiLo [9] studies, were designed to explore the non-inferiority of low activity (1.1 GBq) vs. the standard high activity (3.7 GBq) of RAI and to compare different RRA preparations for each therapeutic arm, both with THW and rhTSH stimulations (for a total of four comparison groups). The results of the ESTIMABL1 trial [8] showed that, at the eight-month evaluation, a low-ablative activity of 1.1 GBq was equivalent to a higher one, irrespective of the treatment preparation (complete ablation of 91.7% vs. 92.9% of rhTSH and THW, respectively). THW provoked a significant deterioration of quality of life [17] and more cases of lachrymal dysfunction when compared with rhTSH stimulation. Similarly, the HiLo study [9] confirmed the equivalence of low 131I activity (1.1 GBq) vs. the former standard of care. On the whole, the ESTIMABL1 [8] and HiLo [9] trials independently support the use of 1.1 GBq of radioiodine under rhTSH stimulation for RRA in low-, but also intermediate-risk DTC. The equivalence of low radioiodine activity for RRA was also confirmed by several meta-analyses [29,30].

However, patients with very large thyroid remnants might require higher iodine activity to achieve a complete remnant ablation. At this regard, Jin et al. [31] showed that an approach in which the activity of RAI therapy is based on Tg values and on RAI uptake on a diagnostic RAI scan can achieve a better ablation rate compared with a fixed activity. This approach might be appropriate for those subjects with huge residual tissue eventually avoiding the need for repeated RAI treatments.

#### Follow-Up Tools for DTC Patients Treated with Surgery Alone

Along with RRA refinement strategies, the available follow-up tools have evolved as well. The main allies of DTC management are neck ultrasound (US), and the dosage of Tg and TgAb. The introduction of highly sensitive Tg (hsTg), whose functional sensitivity is about 0.1 or 0.2 ng/mL, allows for the simplification of the follow-up protocol, avoiding the need for stimulated Tg (stTg). In fact, hsTg proved its equivalence in revealing distant metastases compared with stTg, even in patients without RRA [32,33,34,35].

In patients treated with total thyroidectomy alone (i.e., not followed by RAI), small thyroid remnants might persist resulting in low-detectable Tg [32,36,37]. In these cases, a stable or decreasing trend is associated with remission, while a rising trend is highly suspicious of persistent/recurrent disease [14]. The same consideration can be made for TgAb, as we observe in most cases a spontaneous decline [2,38,39]. Nonetheless, at least half of patients will already have an undetectable Tg after 12 months of follow-up [32,36,37], particularly when surgery is performed in an experienced center. As a consequence, the presence of residual traces of Tg or TgAb should not be considered to be an automatic indication for RAI treatment in lower-risk patients.

In addition, neck US provides an accurate evaluation of thyroid bed and loco-regional lymph nodes (LN), limiting the need for WBS or more invasive imaging [40]. The diagnostic accuracy of neck US coupled with cytology and Tg washout measurement in LN is close to 100% [2,41]. A large proportion of ATA low-risk patients (about 60%) [41] can be classified as being in excellent response after surgery (i.e., post-surgical undetectable hsTg < 0.2 ng/mL or stTg values < 1 ng/mL, without TgAb, and an absence of structural disease at US evaluation). In these cases, RRA does not improve disease outcomes [42].

### 2.2. Adjuvant Treatment

Adjuvant RAI has the purpose to treat (and eventually reveal) microscopic tumor foci, improving disease-free survival. As explained above, the first administration of RAI may encompass both the ablative and adjuvant aims.

#### 2.2.1. Low-Risk Patients

Low-risk patients represent the largest DTC subgroup.

Several retrospective or observational research works have questioned the usefulness of adjuvant RAI in low-risk patients, since no benefit in terms of survival has been proved [10,14,27,28,43].

From this perspective, the updated results of the ESTIMABL1 [44] and HiLo [45] trials found no differences in disease recurrence after RRA within any analyzed subgroup, considering a median follow-up of 5.4 (ESTIMABL1) [44] and 6.5 [45] years, with only the 1.5% and 4.8% rate of abnormal findings for the ESTIMABL1 and HiLo study, respectively. The higher percentage of events in the HiLo study is due to the larger proportion of intermediate-risk patients enrolled in the study (about 40%), [45]. Observational studies with longer follow-up data [46] showed no difference in recurrence rate after 10 years in a DTC series treated for RRA, after both THW and rhTSH stimulation.

On the whole, observational and RCT results suggest not only the effectiveness of the low activity of 1.1 GBq of 131I under rhTSH for RAI therapy, but also the potential futility of adjuvant treatment in a lower-risk setting of DTC patients.

Two RCTs, the ESTIMABL2 Trial [47] in France and IoN trial (NCT01398085) in the United Kingdom, have been designed to explore the effectiveness of RAI vs. no RAI treatment in low-risk patients after surgery. While IoN results are still pending, the ESTIMABL2 results have recently been published [47]. A total of 776 low-risk DTC patients, either pT1am Nx/0 or pT1b Nx/N0, were randomized to postoperative RAI treatment 1.1 GBq under rhTSH stimulation (*N* = 389) or to follow-up (*N* = 387). The study was designed to prove the non-inferiority of follow-up when compared with postoperative RAI treatment in the percentage of patients free from events during the three years following randomization. The event was a composite outcome, including any neck US abnormality (confirmed on cytology or Tg washout), the need for subsequent treatment or serum markers abnormalities (any Tg value above 5 ng/mL, or a Tg on LT4 treatment above 1 ng/mL confirmed on two measurements 6 months apart for the RAI treatment group or a Tg on LT4 treatment above 2 ng/mL confirmed on two measurements 6 months apart for the follow-up group; the appearance of TgAb or an increase of >50% of the Tg Ab titers on two consecutive measurements) [47].

The results of the study were positive and demonstrated no difference in the rate of event-free patients in the two groups (95.6% vs. 95.9% of RAI-treated cases and follow-up groups, respectively 90%CI −2.7–2.2) [47]. Biological events were the more common kind of events and structural (abnormal neck US) or functional (abnormal RAI uptake) events occurred in only eight patients (three in the follow-up group and five in the RAI group). Interestingly, serum Tg above three different cutoffs (0.2, 0.5, and 1ng/mL) was associated with more events at 3 years after randomization, but with a statistically significative difference only of the two latter ones [OR = 3.2 (1.4–7.5) and OR = 5.2 (2.0–13.5)] [47]. Finally, no prognostic value of *BRAF*-mutation was found in the subgroup of the population from which the molecular profile was performed [47]. While waiting for further results from IoN, ESTIMABL2 is the first study to provide strong evidence against the routine use of RAI therapy for low-risk patients, at least on a short-term basis [47].

It is worth noting that with contemporary diagnostic tools, in cases with disease-free status after initial treatment, at least half of the DTC recurrences are observed within 3 years and up to 80% are detected within 5 years of diagnosis [36]. However, the slow-growing attitude of DTC may result in some delayed cancer relapses observed decades after surgery and initial treatments [48,49], and the ESTIMABL2 trial results will need to be confirmed after a longer follow-up of the study population.

#### 2.2.2. Lower-Intermediate Risk Patients

Intermediate-risk DTC represents a very heterogeneous subgroup of DTC, showing variable overlap with low-risk patients in terms of RAI indications and concerns [28,50]. Lower-intermediate risk DTC may be considered as DTC with microscopical extra-thyroidal extension (mETE) and ≤5LN micro-metastases (in the central compartment), but not extrathyroidal invasion, aggressive histology, clinically evident N1 (cN1), lateral neck or mediastinal LN involvement (N1b) [2,51]. For this subset of patients, the risk of recurrence ranges from between 5% and 10%, but it drops to less than 4% if an excellent response is achieved after initial treatment, even in cases with additional risk features [26,52,53]. As a result, the systematic use of adjuvant RAI is controversial in this category [50,53].

The limited utility of RAI in reducing cancer mortality [54] or improving prognosis was observed in several retrospective or observational series [55,56,57]. For instance, Hay et al. [58] analyzed a large cohort of low-risk PTC followed at the Rochester Mayo Clinic, covering a long time span of six decades. The patients were classified as low-risk according to MACIS score <6 (a prognostic score calculated as follows: 3.1 (if aged less than or equal to 39 years) or 0.08 × age (if aged greater than or equal to 40 years), +0.3 × tumor size (in centimeters), +1 (if incompletely resected), +1 (if locally invasive), +3 (if distant metastases present)). It is worth noting that, unlike the ATA risk classification, MACIS is more of a survival rather than a recurrence prognostication tool. This population includes low and intermediate patients according to the ATA risk stratifications [58]. The authors observed no benefit in terms of 20-year mortality and recurrence rates in their population comparing patients undergoing and not undergoing RAI treatment. Interestingly, during the most recent twelve years (1995–2014), the authors found in 740 N1 patients a significantly higher regional recurrence rate for the groups who performed RAI rather than surgery alone (*p* = 0.007) [58]. A possible interpretation of the worse outcome in RAI-treated N1 low-risk patients is the number of metastatic LN on surgical reports: the higher the number of metastatic LN, the higher the probability that patients received RAI (*p* < 0.001), thus indirectly selecting a subset of more aggressive tumors [58]. This same paradox effect was found by other authors and disappears when a propensity matching is performed [10].

Further conflicting data can be found in the literature. A large retrospective analysis by Kim et al. of 8297 intermediate-risk patients [59] failed to prove that RAI reduced the risk of loco-regional recurrence (HR = 0.852, *p* = 0.413), even in cases with additional negative features, such as larger tumor size, multifocality, mETE, lymph node metastases, and *BRAF* mutations. A longitudinal study of 470 PTC patients with mETE N0 or Nx found no significant difference in terms of a structural incomplete response at 1 year in patients treated with or without RAI [60], which was consistent with the results of a retrospective study with a smaller population but a longer follow-up period (median 7 years) [61]. Another study, based on the SEER registry showed some benefits in disease-specific survival, but only in particular subgroups, i.e., male gender (*p* = 0.005), age > 45 years (*p* < 0.001), and larger tumors (*p* = 0.007) [62]. On the other side, Ruel et al. [63] observed a better OS with a 29% chance of decreased risk of death (HR = 0.71; CI 95% 0.62–0.82, *p* = 0.001) in intermediate-risk patients undergoing adjuvant treatment (excluding aggressive variants and multiple primaries) on a large sample of 21 870 DTCs from the National Cancer Database. The authors found the same results even in younger patients (HR = 0.64; CI 95%:0.45–0.92, *p* = 0.016) [63]. Considering RAI activity, lower iodine protocols have been proposed for intermediate-risk patients with mainly lower risk features, but controversial results are available [53,64,65,66,67] (Table 2) and different practices can be observed according to each centers’ experience.

#### 2.2.3. Selective Use of Adjuvant RAI in Low and Lower-Intermediate Risk, According to Ongoing Risk Classification

ATA guidelines suggest taking into account several clinical factors for the decision to administer RAI, such as the biochemical and radiological post-operative status of the patient, the quality of post-operative tools (Tg concentration, and neck US), the experience of the thyroid surgeons, and patients’ opinions.

Based on these premises, evidence from the literature advocates more and more a wait-and-see attitude in low and lower-intermediate-risks DTC, where delayed treatment does not seem to affect the final outcome, while effectively stratifying the recurrence risk and thus reducing the rate of unnecessary RAI [50,51,68,69]. Conversely, emerging structural or biochemical incomplete responses might arouse suspicions even in formerly low-risk patients, supporting additional adjuvant treatment in these selected cases [51,70,71]. The results of the available studies are summarized in Table 3.

The INTERMEDIATE trial (NCT04290663) is an RCT that is enrolling lower-intermediate DTCs that is ongoing in France. The aim is to compare a systematic RAI treatment with 3.7 GBq of ^131^I after rhTSH vs. RAI treatment indication guided by postoperative results (serum Tg values and diagnostic RAI scintigraphy, all patients having normal neck US at study entry). While waiting for higher quality results, the actual evidence is in favor of a careful selection of intermediate-risk DTC to be eligible for adjuvant RAI, notably for those with lower-risk features.

#### 2.2.4. Higher-Intermediate and High Risk

A global agreement is observed in favor of RAI treatment for both higher-intermediate risk and high-risk patients [2,10].

Higher-intermediate risk includes patients with aggressive histology, or extensive lymph node involvement, i.e., >5 metastatic LN, lateral neck metastases, cN1, the presence of ≥3 LN metastases with extranodal extension. In these cases, the risk of recurrence is ≥20% (Table 1).

Considering specific features, some aggressive variants of PTC—i.e., insular, tall cell (TCV), or diffuse sclerosing (DSV) variants—have a recognized worse prognostic role, irrespective of tumor size or other histological factors [72,73]. The five-year survival rate observed was 87.5% and 80.6% for DSV and TCV, respectively, while RAI therapy resulted in effectively reducing mortality (*p* = 0.026) [72]. Insular variant and poorly-differentiated thyroid cancer (PDTC) are rare and aggressive types of DTC, with frequent LN and distant metastatic involvement and a ten-year survival of 50% or less [74]. In these cases, adjuvant RAI therapy is usually recommended even if the benefits remain controversial; some studies have reported some benefits [75,76]. A large retrospective SEER analysis found a longer overall survival after RAI treatment (*p* = 0.001), even without improving the cancer-specific survival (*p* = 0.083) [77].

The preferred RAI activity for these higher-risk classes is usually ≥3.7 GBq, while the treatment preparation mainly relies on THW [16]. However, the most effective protocols are still debated, with several studies showing variable results, according to different preparations and RAI activities.

Considering the timing of RAI administration in higher-risk categories, Yu et al. observed that early RAI therapy (within 3 months) was associated with less biochemical incomplete responses, but it had no influence on the rate of structural disease recurrence [78] while Kim et al. found no such difference within 6 months of surgery [79].

In summary, adjuvant RAI is recommended in all higher-risk patients based on observational evidence demonstrating advantages in recurrences and survival rates. The best RAI administration protocol has not yet been fully identified [80]. Some studies support the higher activity (averagely 3.7–5.55 GBq) of iodine, and THW is the preparation of reference.

### 2.3. Therapeutic RAI

Therapeutic RAI refers to radioiodine administration for persistent/recurrent disease or distant metastases. Loco-regional metastases (i.e., LN metastases and the thyroid bed) are the most frequent sites of recurrence (20–30% of all cases) [81] and, in these cases, patients are often referred for surgery. Distant metastases are rare (<10%) and the most frequently involved organs are the lungs and the bones. RAI therapy is the favored starting approach for metastatic DTC [2]. All metastatic DTC cases should be discussed in a multidisciplinary tumor board. The presence of bulky or threatening metastatic sites should lead us to consider focal treatments (e.g., surgery, interventional radiology, and radiotherapy) as an alternative, or in association with RAI therapy, as well as in the case of single or oligo metastatic disease, in order to limit repeated RAI administrations.

Repeated therapeutic activities are administrated (every 3, 6, or 12 months) until a complete response is obtained or refractory disease occurs [50,82,83]. The 10-year disease-specific survival is 92% for those patients who show uptake and respond to RAI treatment, while it drops to 30% and 10% for the patients who do not achieve a complete response to RAI despite uptake and those who do not uptake RAI, respectively [84]. Predictors of a good response to RAI usually are: younger age, iodine avid metastases, a small metastatic disease burden, well-differentiated histology, and lower Tg values at the first RAI treatment [81,84,85,86,87,88,89]. In contrast, the presence of PDTC, bone metastases, macronodular lung lesions, and the absence of RAI uptake are independent predictors of RAI failure and worse prognosis [86,90,91].

Unfortunately, even in the presence of significative RAI uptake, disease progression can occur in favor of RAI refractory disease, as observed in follicular thyroid carcinoma (FTC) [81], and RAS mutated tumors [92].

Miliary lung metastases, sometimes evident only on WBS without radiologic evidence of disease, along with younger age groups (<40 years), have a better response after RAI therapies, allowing for a 10-year survival rate of 90.9% vs. 68.9% and 30.6% for patients with detectable subcentimetric or larger metastases, respectively [93].

Bone metastases are the second-most-frequent metastatic site. Up to 13% of patients will develop bone lesions [94] and they represent a negative prognostic factor, reducing patients’ overall survival and quality of life, due to pain or skeletal complications [86,94,95]. In almost 32% of cases, patients may achieve a complete response with a combination of RAI and loco-regional treatments [96,97], which significantly improves the overall survival (7.7 years vs. 3.9 years, for combined therapies and RAI alone, respectively) [96]. However, according to a multicenter survey, up to 37.1% of patients will show harmful or fatal skeletal events [98]. Favorable prognostic factors are younger age (<45 years old) and less than three metastases [94]. In particular, cases where only RAI uptake at WBS scans is detectable, without corresponding lesions on conventional radiologic studies, have the highest chance of a complete response [95,99].

In the event of metastatic disease, higher RAI activity (≥3.7 GBq) is necessary to produce a therapeutic effect and two approaches can be used: fixed doses or dosimetry protocols. An empiric fixed RAI activity between 3.7–7.4 GBq is the most frequently used strategy, as this approach is easier and represents a good compromise in terms of effectiveness and adverse events. However, empiric doses ignore the individual radioiodine kinetics and safety concerns have been raised as fixed activities often exceed the maximum tolerated dose in older subjects, even in cases with normal renal function [100]. On the other hand, the dosimetry approach is based on the assessment of the lesions’ radioiodine uptake and bone marrow tolerance, respecting the “as high as safely administrable” rule [101]. Dosimetry is supposed to provide advantages in several situations: from limited loco-regional disease to advanced stages, where tumor burden can be measured by conventional radiologic tools; in younger patients, where lower activities may be equally effective, as well as desirable; or also in disseminated lung metastases, in order to prevent lung fibrosis [101]. However, dosimetry requires more time and resources, along with specific skills and the administration of low ^124^Iodine activities to test lesions’ iodine metabolism, which could also result in stunning the metastases, reducing the final effectiveness of the therapeutic RAI [101,102,103]. It is worth noting that the activity administered at each RAI course is usually greater with the dosimetry approach than with the empiric approach. Despite the theoretical advantages of dosimetry, Deandreis et al. [104] found no differences in terms of survival, comparing patients who underwent fixed doses (3.7 GBq) or whole-body dosimetry in two major oncological centers. No difference in survival was found after stratifying for the major predictive factors of RAI response (age and metastatic burden) [104]. Interestingly, the patients treated with the dosimetric approach received larger cumulative activities than those treated with fixed activity [cumulative activity 14.8 GBq (range, 1.8–52.5 GBq) at Gustave Roussy vs. 24.2 GBq (range, 2.7–112 GBq) at Memorial Sloan-Kettering Cancer Center (*p* < 0.01)].

Another issue concerns hormonal preparation, i.e., rhTSH or THW. THW results in a higher uptake and a longer half-life in the metastatic lesions of RAI [105] and can also reveal more metastatic foci [106].

However, due to the well-known drawbacks and potential health risks of THW, rhTSH has been explored as an alternative protocol for metastatic patients [107]. Some studies showed benefits and the same effectiveness of rhTSH preparation compared with THW [108,109,110,111,112,113,114,115]. However, most of these cohorts are small and retrospective, and the follow-up of the patients treated with rhTSH is most often significantly shorter than that of the patients treated after THW, preventing any reliable conclusions (Table 4). A systematic review confirmed rhTSH and THW equivalence for ablative purposes, while the authors found insufficient data to conclude the same in a metastatic framework [116]. THW is still the standard preparation for therapeutic RAI administration for patients with distant metastases and rhTSH should be considered to be an alternative option only in rare cases in which THW is contraindicated.

Figure 1 summarizes the indications for RAI therapy considering both dynamic risk assessment and ^18^FDG-PET information.

## 3. Pediatric DTC

DTC in the pediatric and adolescent populations represents a rare disease with a higher rate of local and distant metastatic disease compared with adult DTCs [117]. Despite the aggressive presentation [117,118,119,120,121,122], pediatric DTCs are the most RAI-sensitive cancers, and a cure can be achieved in up to 52% of the cases [121]. These young patients have similar recurrence rates but lower specific mortality rates than adults in studies with long follow-up periods [117,118,119,120,122], and the rare cases of cancer-related deaths occur at an adult age [117].

Overall, RAI treatment after bilateral thyroid surgery is associated with an improved outcome [123]. The activity of RAI to be administered is calculated according to the body weight of the young patient (1–1.5 mCi/Kg) and the preferred preparation method is THW, although some experiences with rhTSH report non-inferior results [124]. Multiple RAI treatment courses are often necessary to obtain complete disease remission, and in a retrospective analysis of 125 children, 22% of patients required three or more RAI courses [120]. Considering metastatic cases, the cumulative RAI activity ranges from an average of 11.98 GBq to a maximum activity of 22.2 GBq, according to surveys [121,125], although Pires et al. found no therapeutic benefit after 14.8 GBq of iodine administration in their cohort of 118 patients [126]. Interestingly, some delayed responses to RAI were observed in pediatric patients with lung metastases on simple surveillance after repeated RAI treatments [127] supporting a less frequent RAI treatment schedule in these patients. Due to the more aggressive therapeutic approach applied to pediatric DTC, several concerns about iodine safety have arisen over time, notably regarding fertility and additional oncological risks.

## 4. Side Effects of RAI

Even if RAI therapy is universally considered a safe and well-tolerated treatment [14], some side effects are possible. In fact, in addition to the thyroid gland, the presence of sodium/iodine symporter (NIS) has been shown in several organs, allowing significant iodine uptake, but also leading to an increased susceptibility to adverse events (AEs) during RAI treatments [128]. The most frequently involved non-thyroidal tissues are the salivary and nasolacrimal glands, but potential toxicity on gastrointestinal tissues, bone marrow, and gonads has also been described during the course of succeeding radioiodine administrations [128,129,130,131,132,133]. The real incidence of RAI-related AEs varies, according to the studies and the detection methods. The risk of any RAI-related toxicities increases with the increasing cumulative administered activity of RAI [134,135,136], even if few reliable data are available about the rate of AEs after very high RAI doses. However, potential AEs may develop anyway, even after one RAI treatment or low activity administration, according to individual susceptibility.

The most common chronic side effects are summarized in Table 5.

Although extremely rare, the onset of RAI-related second malignancies represents a controversial and feared late AE. Growing concerns have risen especially for younger DTC patients, due to the more aggressive therapeutic approaches applied and the long life expectancy. A study based on a SEER large sample of 27050 pediatric DTC patients showed that the risk of second neoplasms appeared concrete for both hematologic (RR = 1.51; CI 95: 1.08 to 2.01)—including leukemia (RR = 1.92; CI 95:1.04 to 3.56)—and solid cancers (RR = 1.47; CI 95: 1.24 to 1.74)—including breast cancer (RR = 1.46; CI 95: 1.10 to 1.95) [157]. Younger DTC survivors of ≥20 years disclosed the greater risks of secondary neoplasms, and RAI-related cancers were estimated to be 6% and 14% for solid and hematologic tumors, respectively [157]. Another SEER-based study showed a higher rate of second breast cancer in younger DTC patients who performed RAI therapy compared with both non-RAI-treated patients (HR 1.65; CI 95: 1.33–2.05, *p* < 0.001) and the general population (HR 1.21; CI 95:1.02–1.44, *p* < 0.05) [158]. However, breast cancer risk remains controversial and the above results have not been confirmed in a large and focused metanalysis of 200247 DTC patients treated with or without radioiodine [159]. Similarly, Kim et al. [160] performed a propensity-score analysis on a wide retrospective multicenter sample of 24318 patients, including RAI-treated and non-RAI-treated subjects, concluding that there was an absence of significant risk of the occurrence of second tumors.

Considering the high rate of salivary gland AEs, several preventive strategies have been investigated in pre-clinical and clinical settings [131,161]. Pre-clinical studies explored the efficacy of various agents, including antioxidants (i.e., Vitamin E) and nutraceutics (i.e., *Curcuma longa*, *Ocimum sanctum*, and zinc), showing potential benefits in non-human subject tests [161]. A clinical trial analyzing the influence of Vitamin C on salivary glands’ iodine absorption found only limited effects, irrespective of the moment of administration during RAI therapy [162]. Similarly, pilocarpine, a parasympathomimetic drug stimulating salivary glands, did not prove to provide any substantial advantages, with potential severe toxicities in selected categories, i.e., patients with asthma or cardiac diseases [161]. Amifostine, a supposed salivary gland radioprotector, showed potential effectiveness in two clinical trials [163,164], but failed to confirm the same results in a systematic review [116], and its tolerance and costs further limited its application in clinical practice [161]. The potential benefits of sialagogues agents, such as lemon candies and lemon juices, have been reported thanks to the increase in salivary function and, thus, in iodine washout [161]. However, some concerns emerged after evidence of a higher rate of AEs in early lemon candy eaters (<24 h after iodine administration), due to the concurrent rise in blood flow to the glands, which potentially increased iodine uptake and retention [165]. This event, also named the “rebound effect”, has not been definitely proven [166,167], but the timing of sialagogue administration remains controversial, as well as potential rebounds during the first 24 h after iodine delivery [161,166,168].

## 5. RAI Refractoriness

### 5.1. Definition and Prognostic Factors

Iodine refractoriness implies the loss of effective radioiodine uptake and the absence of a therapeutic effect of RAI treatment. The clinical evidence of refractory disease corresponds to a biological tumor change, due to the critical reduction or the loss of specific iodine transporters and other fundamental proteins involved in iodine metabolism.

According to the ATA guidelines, RAI refractoriness is defined by: (i) the absence of any RAI uptake in known metastatic sites; (ii) the loss of RAI uptake after former evidence of RAI uptake; (iii) the presence of a heterogeneous uptake with the presence of RAI-avid and non-RAI-avid metastases; or (iv) the evidence of disease progression after a significative RAI uptake within 6 to 12 months of the last RAI treatment [2]. In line with this definition, cases labeled as RAI refractory should not undergo further treatment, due to their ineffectiveness [2]. Several experts have tried to refine the definition of RAI refractoriness, proposing additional features [169], such as the evidence of ^18^FDG-PET avid-metastases [170] or a cumulative dose exceeding an activity of 22.2 GBq (600 mCi) [169]. Another scenario that can be assimilated with that of RAI refractory disease is that of locally advanced and unresectable DTCs; these patients have an overall survival similar to RAI refractory patients and should be treated accordingly [171].

Overall, patients fulfilling the definition of RAI refractory disease will not achieve cure with RAI treatment alone but sometimes can obtain some benefit from further RAI, and these cases should be discussed in a case-by-case fashion in expert multidisciplinary boards.

In the event of discordant RAI-avid and RAI refractory metastases, patients formally fit the definition of refractoriness, but they may still achieve some benefits from a combination of RAI and other therapeutic strategies [169]. It is worth noting that the overall prognosis of the patient is driven by the less-differentiated foci of disease (i.e., those that do not uptake RAI) and the treatment of these sites should be prioritized. Even if ^18^FDG-PET uptake is associated with worse OS, regardless of the RAI uptake status [172,173], some responses to RAI can be observed in patients with distant metastases that show both ^18^FDG-PET and RAI uptake [174].

Several clinical and molecular features have been associated with a worse outcome in DTC patients and the emergence of RAI refractoriness. Considering clinical features, older age has been traditionally associated with a worse DTC prognosis, since the elderly carry an increased risk of metastatic disease [175] and of refractory DTC [84]. In RAI refractory patients, an age of >60 years-old (HR = 8.498, CI 95:1.555–46.427, *p* = 0.0135) and the male gender (HR = 5.435, CI 95:1.261–23.256, *p* = 0.0231) negatively affected patients’ survival [176]. However, Saie et al. [177] observed a reduction in the overall survival (4.65 years, 95% CI: 2.04–5.68, *p* = 0.0008) only in patients older than 75 years, while they failed to prove an association between age and progression-free survival in their cohort. Other reports did not confirm the negative prognostic role of age in PDTCs, even considering different thresholds [178].

Luo et al. [179] found that the presence of gross ETE, aggressive histology, BRAF-V600E, and telomerase reverse transcriptase (TERT) promoter mutations were related to the occurrence of refractory DTC. As stated above, aggressive histotypes usually develop distant metastases and develop RAI refractoriness [73,81]. Histology is also associated with RAI uptake, with FTC and Hürtle cell carcinoma (HCC) showing opposing abilities in iodine concentration, but the same higher rate of progression after RAI [81]. Patients with a gross disease burden, with larger pulmonary or bone metastases, usually do not benefit from RAI treatment and most often have refractory phenotypes [91,93,95].

Considering the molecular signature of refractory DTC, the mitogen-activated protein kinase (MAPK) pathway—mainly BRAF mutations and the TERT promoter molecular alterations—is most frequently involved in DTC tumorigenesis and development of RAI refractoriness [180]. The most frequent mutation in PTCs is BRAF-V600E, found in up to 60% of PTC [181]. The BRAF mutation causes the uncontrolled activation of the MAPK pathway and is associated with a more aggressive phenotype and a less favorable outcome [182]. Moreover, BRAF mutated tumors have a less-differentiated phenotype with the downregulation of iodine metabolism genes, such as NIS and thyroperoxidase (TPO) [183,184] resulting in a decreased or absent radioiodine uptake and the loss of RAI sensitivity [92]. TERT promoter mutations are considered to be a late event in DTC carcinogenesis and are often found in more aggressive histological types, such as PDTC [185] and anaplastic thyroid cancer [186]. The BRAF and TERT duet is associated with a poor outcome [182,187].

Refractory tumors have an enhanced expression of glucose transporters, such as GLUT1 [188,189]. This change is reflected in the occurrence of ^18^FDG-PET avid-metastases and is associated with iodine refractoriness, aggressive disease, and poor outcome [170,190,191].

The role of oxidative stress is also recognized in thyroid cancer carcinogenesis and RAI refractoriness [192]. Indeed, the redox status of the cells influences the expression of the NIS [193]. In BRAF-mutated tumors, an increased expression of NOX4 is observed via a TGF-beta-SMAD3-dependent pathway and plays a role in NIS repression [194]. This phenomenon might be due to epigenetic modifications as it has been demonstrated to be reversible [192].

### 5.2. Strategies to Overcome RAI-Refractoriness

As stated above, the RAI refractory phenotype implies the loss of fundamental cellular proteins involved in iodine uptake and metabolism. Over recent decades, growing attention to the molecular landscape of refractory tumors has driven research into mechanisms underlying refractoriness, allowing initial attempts to restore iodine sensitivity, with variable results. The earliest molecular pathways explored in this framework have been those of retinoid acid (RA) receptors and peroxisome proliferator-activated receptor ɣ (PPAR ɣ), but with disappointing results [195].

The main breakthrough in this framework was provided by the finding of MAPK signaling as the pathway that was the most frequently involved in the refractoriness process [196,197]. The major trigger of the MAPK pathway is represented by BRAF-V600E [181,185,197], a leading mutation that, by activating the downstream kinases MEK and ERK, produces an uncontrolled stimulation, resulting in a downregulation of iodine metabolism genes, such as NIS and TPO [183,184,196,197,198]. The more recent attempts to stimulate the redifferentiation process have been built around these molecular steps, finding through the selective BRAF and MEK inhibitors the pivotal therapeutic agents [196,197]. The first anti-MEK agent used in clinical practice was selumetinib, which was administrated to a cohort of 24 iodine refractory patients with variable molecular status (RAS, BRAF mutated, or wild-type) [199]. Of the 20 evaluable subjects, five had a partial RECIST response and three had a stable disease, with RAS-mutated patients being better responders [199]. During the following years, other studies focused on specific BRAF inhibitors and on the combination of BRAF and MEK inhibitors. In the former case, small cohorts of BRAF-V600E mutated refractory DTC [200,201,202] have been treated with dabrafenib or vemurafenib alone, finding a rate of objective response ranging from 20 to 50%. The results of these series are listed in Table 6.

A histological drug-induced change was evidenced in a patient harboring the BRAF-K601E mutation and treated for 8 weeks with trametinib ± dabrafenib [207]. In this case, the biopsy following the target therapies treatment showed a more differentiated pattern and a reduced mitotic rate, compared to the initial histology. Moreover, the patient developed overt hyperthyroidism and a metabolic response on ^18^FDG-PET [207].

Further insights into molecular changes during redifferentiation therapies have emerged over time. As previously reported [208], the inhibition of the MAPK pathway by anti-ERK agents induced the overexpression of the HER3 receptor, which reduced the effects of these drugs. Lapatinib, a HER3 inhibitor, proved to prevent this mechanism, improving the MAPK inhibitor effectiveness and paving the way for this therapeutic combination. A pilot study by Tchekmedyian et al. [206] explored the combination of vemurafenib and a monoclonal antibody anti-HER3 (CDX-3379), due to the supposed reduction of the anti-BRAF inhibition provided by the rebound of tyrosine kinase erbB-3 (HER3). The authors [206] observed two partial responses (PR) and two progressive diseases (PD) in a series of six BRAF-V600E-mutated patients and confirmed the interest of further investigations, including these combinations [206].

Considering the promising results of MAPK inhibitors, several trials are ongoing worldwide, in order to find an effective redifferentiation protocol. MERAIODE (NCT03244956) is a multicenter interventional trial, promoted by the French network TUTHYREF, and aimed to explore the effectiveness of trametinib (anti-MEK) alone in the event of RAS-mutated DTC or a combination of dabrafenib (anti-BRAF) and trametinib, in case of BRAF-V600E-mutated DTC. After the differentiation protocol, patients with restored iodine sensitivity undergo RAI treatment at the fixed activity of 5.55 Gbq after rhTSH stimulation. Preliminary results in 21 BRAF-mutated patients showed PR in 38% (95% CI 18–61), SD in 52% (95% CI 30–74) and PD in 10% (95% CI 1–30). The tumor control rate was 90% and objective responses were observed in 38% of cases, with no patients experiencing adverse events of grade 4 or 5 [205]. The preliminary results of the 10 evaluable patients of the RAS cohort were less promising with only two PR and seven SD [209].

Other trials with similar interventions from Japan (NCT04554680) and the United States (NCT02152995) are actually ongoing in BRAF-V600E or RAS-mutated DTC patients. The former is aimed at exploring the combination of dabrafenib and trametinib after one or four weeks of therapy in a five-subject population; the latter explores the anti-MEK trametinib after four weeks of therapy, in a cohort of thirty-four patients. The results of these studies are pending.

## 6. Concluding Remarks

Radioiodine represents the oldest therapeutic approach for DTC and the only one capable of obtaining a complete remission at the metastatic stage. However, despite the long tradition of its use, RAI strategies still have several unsolved issues.

Indications for RAI therapy mostly rely on the ATA recurrence risk stratification, which provides an estimate of the risk of persistent or recurrent disease after thyroid cancer surgery. This system is complex due to the numerous items to be taken into account and can be hard to apply in real life practice. To achieve a fine-tuned selection for ablative, adjuvant or therapeutic RAI and to overcome the present limitations of the ATA stratifications, ongoing assessments with serum Tg and TgAb assessment and neck US may be considered. Furthermore, RAI protocols and timing are not standardized, resulting in a variety of behaviors according to single-center practice. In this light, focused studies are needed to fill these gaps of evidence and improve our practice.

An improved risk-based selection of patients and the growing availability of high-quality data have led to a more accurate definition of candidates for ablative RAI and the most effective protocols for lower-risk patients. In most cases, very low and low-risk patients may safely avoid systematic RAI therapy, and eventually be selected for a low-dose protocol, as supported by the ESTIMABLE1, HiLO, and ESTIMABLE2 RCTs [8,9,47]. The intermediate-risk category still appears to be too heterogeneous to draw univocal and comprehensive indications for RAI treatment. Splitting this category into lower- and higher-intermediate-risk might help to refine the therapeutic indications in this group and a post-surgical workup is likely to improve RAI decisions. The results of the IoN and Intermediate RCT will probably shed light on these issues.

Adjuvant RAI is performed in patients who do not clearly show any evidence of disease, and so, a variable proportion of them are likely to be overtreated. In this light, high-risk patients are good candidates for adjuvant RAI, but a tailored approach is required for most of the other risk classes, where the recurrences are rarer [2,16]. On the other hand, the high-risk class still lacks high-quality evidence, which is partially ascribed to the rarity of these patients and the optimal treatment schedule is yet to be established.

Some safety issues are still awaiting clarification. Prevention strategies are urgently needed for frequent RAI AEs, such as xerostomia and salivary gland impairment. On the other hand, due to the unclear risks of RAI-related second cancers, further insights are required and greater caution should be exercised regarding the long-term safety of iodine.

Finally, despite broad progress in terms of knowledge about and protocols for tumor redifferentiation, several issues are still unresolved. Patient selection still needs to be fine-tuned, along with the whole molecular pathway underlying the refractory phenotype. Further molecular alterations and relative targetable drugs should be explored and larger randomized trials are needed. Moreover, the optimal schedule for redifferentiation and the ideal sequence with other available therapeutic options, such as tyrosine-kinase inhibitors, has yet to be established.

In conclusion, RAI therapy still represents a cornerstone of DTC treatment. A tailored approach according to the risk features of patients and their tumors has led to a progressive refinement of therapeutic strategies, optimizing RAI indications and patients selection. Several questions remain unanswered and high-quality studies are needed to further clarify the future RAI strategies.

## Figures and Tables

**Figure 1 cancers-14-03800-f001:**
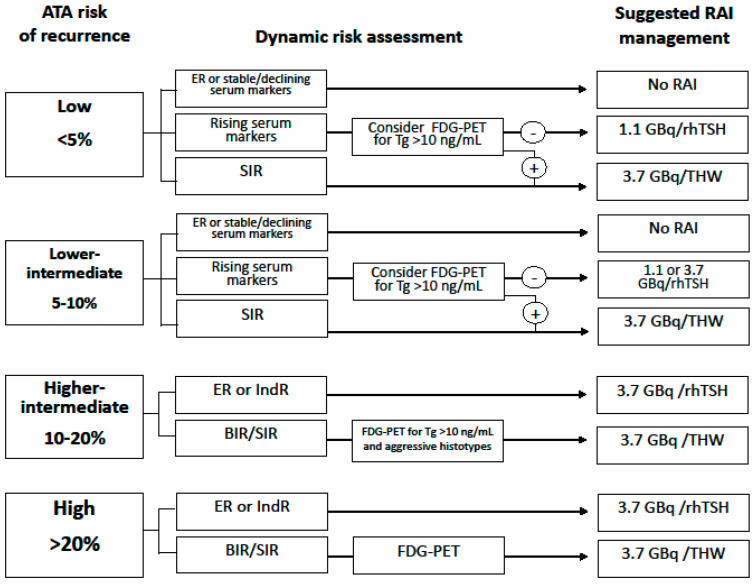
Indications for RAI therapy considering both dynamic risk assessment and ^18^FDG-PET information. Abbreviations: RAI, radioiodine; ATA, American Thyroid Association; FDG-PET, fluorodeoxyglucose-positron emission tomography; Tg, thyroglobulin; rhTSH, human recombinant thyroid stimulating hormone; THW, thyroid hormone withdrawal; ER, excellent response; SIR, structural incomplete response; BIR, biochemical incomplete response; IndR, indeterminate response.

**Table 1 cancers-14-03800-t001:** AJCC TNM Staging 8th edition [12] and ATA risk classification and RAI recommendation according to ATA 2015 [2].

T	N	M	Additional Features	Stage ≥55 Year	Stage <55 Year	Risk of Death (%)		2015 ATA Risk	Risk of Recurrence (%)	RAIT Recommended
						≥55 Year	<55 Year			
1a	0	0	-	I	I	<2	<2	L	<5	No
1–2	0	0	Uni or Multifocal *	I	I	<2	<2	L	<5	Not routine
1–3	0	0	FTC minimal vascular invasion	I/II	I	2–5	<2	L	<5	Not routine
3	0	0	-	II	I	~5	<2	L	<5	Not routine
1–3	1a	0	≤5 microscopic N1 (<2 mm)	II	I	~5	<2	L	<5	Not routine
1–3	0	0	Minimal ETE	I/II	I	2–5	<2	I	5–20	Favored (consider size)
1–3	1a/b	0	>5 N1 of <3 cm	II	I	~5	<2	I	5–20	Favored
1b–3	0	0	*BRAF* mutation	I/II	I	2–5	<2	I	5–20	Favored
1–3	any	0	Aggressive histology *	I/II	I	2–5	<2	I	5–20	Favored
1–3	any	0	Uptake outside thyroid bed on RxWBS	I/II	I	2–5	<2	I	5–20	Favored
1–3	any	0	Vascular invasion *	I/II	I	2–5	<2	I	5–20	Favored
1–3	any	0	FTC > 4 foci of vascular invasion	I/II	I	2–5	<2	H	>20	Yes
1–3	1a/b	0	N1 > 3 cm	II	I	~5	<2	H	20	Yes
4a	any	0	-	III	I	5–20	<2	H	20	Yes
4b	any	0		IVa	I	>50	<2	H	20	Yes
Any T	Any N	1	-	IVb	II	>80	~5	H	20	Yes
Any T	Any N	0	Incomplete tumor resection	-	I		<2	H	20	Yes
Any T	Any N	0	Tg out of proportion with RxWBS findings	-	I		<2	H	20	-

* Aggressive histology (e.g., tall cell, columnar, insular, and poorly differentiated), vascular invasion and multifocal foci in combination with size, lymph node status, and age can increase the risk of the patient and may be an argument for radioiodine remnant ablation for the ATA 2009 guidelines. Acronyms: AJCC, American Joint Committee on Cancer; ATA, American Thyroid Association; yr, year; RAIT, radioiodine treatment; T, tumor; N, node; M, metastasis; L, low-risk; I, intermediate risk; H, high risk; FTC, follicular thyroid carcinoma; ETE, extrathyroidal extension; RxWBS: post-therapeutic whole-body scan; N1, positive lymph nodes; Tg, thyroglobulin.

**Table 2 cancers-14-03800-t002:** Overview of studies considering lower RAI activity in intermediate-risk patients with mainly lower risk features.

Authors, Reference	Study Design	Intermediate-Risk (N/Total)	THW Preparation (%)	Iodine Activity-GBq (N)	Median Follow-Up [Range]	Main Outcomes
**Welsh et al.** [64]	Prospective	53/53	100	1.1 (53)	24 years (4–34)	51% of unsuccessfully ablated patients; 30 years DSS and OS of 87% and 62%, respectively, in unsuccessfully ablated patients, without significant differences between groups.
**Rosario et al.** [53]	Retrospective	152/152	72.4	1.1 (152)	76 months (18–140)	Persistent/recurrent disease in 6% of patients.
**Han et al.** [65]	Retrospective	176/176	100	1.1 (96) vs. 5.5 (80)	7.2 years (3.3–9.4)	No significant differences in BIR/SIR between high vs. low RAI activity groups.
**Jeong et al.** [66]	Retrospective	204/204	100	1.1 (80) vs. 3.7–5.5 (124)	10 years (NA)	BIR/SIR 10.5% vs. 25% in high vs. low RAI activity groups, respectively (*p* = 0.01); Need for additional RAI in 6% vs. 22% of high vs. low RAI activity groups, respectively (*p* = 0.001).
**Gomez-Perez et al.** [67]	Retrospective	47/174	-	1.1 (13) vs. ≥1.1 (34)	-	Recurrent disease for 67% vs. 24% of low vs. high RAI activity groups, respectively (*p* = 0.003).

Abbreviations: N, number; THW, thyroid hormonal withdrawal; DSS, disease-specific survival; OS, overall survival; NA, not available; BIR, biochemical indeterminate response; SIR, structural indeterminate response; RAI, radioiodine.

**Table 3 cancers-14-03800-t003:** Selected use of RAI in intermediate-risk patients after ongoing risk stratifications.

Authors, Reference	Study Design	Population (N)	ATA-Risk Patients (N)	Group Comparison (N)	Follow-Up (Range/SD)	Disease Recurrences
**Ballal et al.** [70]	Retrospective	254	IR (254)	(A) surgically ablated (125) (B) non-surgically ablated ^a^ (129)	median 10.3 years (1–21)	No significant differences
**Grani, Lamartina et al.** [51]	Retrospective	252	LR (204) IR (68)	(A) Cohort 1 (116): TT and RAI (B) Cohort 2 (156): TT and DRS ^b^	(A) median 8 years (3–12) (B) median 4 years (3–6)	No significant differences
**Abelleira et al.** [71]	Retrospective	307	LR (191) IR (116)	(A) Low-dynamic ^c^ LR+ IR (166) (B) High-dynamic ^c^ LR+IR (141)	(A) and (B) mean 59.5 months (±22.31)	SIR for LR: A (2%) vs. B (5%), *p* = 0.3 SIR for IR: A (5%) vs. B (22%) *p* = 0.008

a, non-surgically ablated patients showed evidence of disease at diagnostic whole-body scan and were referred to RAI; c, dynamic risk stratification has been assessed after initial treatment: patient at low-risk didn’t undergo RAI; b, after total thyroidectomy the decision to perform RAI was deferred for around 12 months for appropriate DRS. Abbreviations: N, number; SD; standard deviation; SIR, Structural incomplete response; LR, low-risk patients; IR, intermediate-risk; THW, thyroid hormonal withdrawal; TT, total thyroidectomy; RAI, radioiodine therapy; DRS, dynamic risk stratification.

**Table 4 cancers-14-03800-t004:** Overview of studies exploring rhTSH preparation for therapeutic radioiodine treatment in patients with distant metastases.

Authors, Reference	Study Design	Number of Patients (rhTSH/Total)	Median Age (Years) [Range] rhTSH	Aggressive Histology ^a^ (%)	Type of RAI Protocol	Metastatic Sites	Median Follow-Up [Range/SD]	Response (N)	OS Difference with THW
**Lippi et al.** [108]	Retrospective	12/12	- [48–75]	83.3	Dosimetry(100%)	lung, bones, and other ^b^	12 months [-]	Biochemical ^c^ (10) Tg reduction 40%; Tg stability 20%; Tg increase 40%	NA
**De Keizer et al.** [109]	Prospective	16/16	73.1 [41–87]	68.7	Empiric doses (100%)	lung, bones, and other ^b^	3 months [-]	Biochemical ^c^ (11) Tg reduction 27%; Tg stability 18%; Tg increase 55%	NA
**Tala et al.** [110]	Retrospective	58/175 (82 THW and rhTSH)	60 [20–89] ^d^	63.8 ^c^	Dosimetry (100%)	lung and/or bones	3.4 years [1.3–10.3] ^d^	Structural (43) ^d^ CR 19%; PR 0%; SD 35%; PD 46%	No difference
**Zagar et al.** [112]	Prospective	18/18	72 [37–83]	77.8	Empiric doses (100%)	lung, bones, and other ^b^	50 months [15–19]	Biochemical ^c^ (18) Tg reduction 17%; Tg stability 22%; Tg increase 61%	NA
**Klubo-Gwiezdzinska et al.** [111]	Retrospective	15/56	62.4 ^e^ [±12.6]	35.7	Dosimetry (80%)	lung, bones, and other ^b^	72 months ^e^ [±36.2]	Structural (15) CR 7%; PR 0%; SD 73%; PD 20%	No difference
**Rani et al.** [113]	Prospective	37/37	48.7 [14–70]	24.3	Dosimetry (100%)	lung and/or bones	-	-	-
**Simoes Pereira et al.** [81]	Retrospective	68/95	65.5 [22–85] d	11.8	Empiric doses (100%)	lung, bones, and other ^b^	82 months [8–332]	Structural (67) ^d^ CR 6%; PR 4%; SD 30%; PD 60%	No difference
**Gomes-Lima et al.** [115]	Retrospective	27/55	59 [47.5–65.5] d	30.0	Dosimetry (89%)	lung, bones, and other ^b^	4.2 years [3.3–5.5] ^d^	Structural (27) CR 0%; PR 63%; SD 11%; PD 56%	No difference
**Tsai et al.** [114]	Retrospective	37/88	46.1 [-]	0	Empiric doses (100%)	lung, bones, and other ^b^	6.5 years [1.0–18.1]	-	No difference

a, including follicular thyroid cancer, Hürtle cell, poorly differentiated thyroid cancer, or aggressive histotypes (i.e., tall cell variant). b, including neck recurrences and/or liver, brain, or other rare metastatic sites. c, Biochemical response refers to the change of Tg levels after therapy. d, data refers to the group treated exclusively with rhTSH. e, age and follow-up are expressed as mean (standard deviations). Abbreviations: rhTSH recombinant human TSH; SD, standard deviations; RAI, radioiodine; N, number; OS, overall survival; CR, complete response; PR, partial response; SD, stable disease; PD, progressive disease; NA, not applicable.

**Table 5 cancers-14-03800-t005:** Summary of radioiodine side effects.

Site	Description	Frequency (%)	Activity (GBq)	References	Commentary
Eye	Inflammation of the lacrimal gland and xerophthalmia	16 (92% at least one altered lacrimal test)	2.96–22.2	[131,135,137,138,139,140,141,142]	Test alteration is not related to patient’s symptoms.
	Obstruction of lacrimal duct and epiphora	2.2–18	>5.55		
	Conjunctivitis (chronic or recurrent)	23	3.7–70.3		
Salivary glands	Sialadenitis:			[137,142,143,144,145,146]	Linear correlation to cumulative activity, more than half of patients develop xerostomia even in the absence of acute post-treatment symptoms. 5% of xerostomia with 1.5 GBq
	- acute	2–67	3.7–48.1		
	- chronic (xerostomia, obstruction)	2–43	1.48–48.1		
	Atrophy	21–78	3.7–7.4		
Taste and Smell	Transient loss or change in taste and smell	2–58	1.48–48.1	[137,143,144]	Dependent on administered activity.
Nose	Pain Epistaxis	Rare	> 7.4	[144]	
Thyroid	Radiation thyroiditis		>2.8	[147]	
	- total thyroidectomy without large remnants	Rare			
	- lobectomy	60			
Gastrointestinal system	Nausea	5–67	1.48–16.5	[144,145,148]	Correlation with administered activity. No symptoms with an activity of 1.1 GBq or less. Nausea starting from 1.5 GBq. Vomiting 1% with <3.7 GBq.
	Vomiting	1–15	3.7–16.5		
Bone marrow	Any hematological abnormality	1—100	3.7–38.5	[137,144,149]	Risk increases with cumulative dose and frequency of treatments. Grade > 3 abnormalities are rare.
Fertility	Transient ovarian failure	8	1.1–40.7	[27,150]	Consider cryopreservation if repeated treatments are necessary or activities higher than 3.7 GBq are required in fertile men.
	Transient or permanent testicular failure	100	1.1–49.4		
	Prolonged or permanent hormonal impairment (FSH increase)	81	>22		
Second Malignancy	Solid cancer and leukemia	Rare	>7.4	[151,152,153,154]	Linear correlation to dose. +27% increase in risk compared to general population.
Lung	Pulmonary fibrosis	Rare	21–71	[155,156]	Usually pediatric DTC patients with lung metastasis; increased risk after several consecutive RAI courses and higher cumulative activity.

Abbreviations: DTC, differentiated thyroid cancer; RAI, radioiodine.

**Table 6 cancers-14-03800-t006:** Overview of published or ongoing studies on redifferentiation with anti-MEK or anti-BRAF drugs.

Authors/Identifier, Reference	Drug	Patients (N)	Molecular Findings (N)	Restored RAI Uptake (N)	Complete Response [N (%)]	Partial Response [N (%)]
**Ho et al.** [199]	Selumetinib + ^131^I	24	*BRAF-V600E* (9) *NRAS* (5) *RET/PTC* (3) WT (3)	8	0	5 (25)
**Rothenberg et al.** [200]	Dabrafenib + ^131^I	10	*BRAF-V600E* (10)	6	0	2 (20)
**Jaber et al.** [203]	anti-MEK and/or anti-BRAF + ^131^I	13	*BRAF-V600E* (9) *NRAS/KRAS* (3) WT (1)	9	0	0 (0) ^a^
**Dunn et al.** [202]	Vemurafenib + ^131^I	12	*BRAF-V600E* (10)	4	0	4 (25)
**Iravani et al.** [204]	anti-MEK ± anti-BRAF + ^131^I	6	*BRAF-V600E* (3) *NRAS* (3)	4	0	3 (50)
**Leboulleux et al.** [205]	Trametinib + dabrafenib + ^131^I Trametinib	21 10	*BRAF-V600E (21)* *RAS (10)*	20 6	0 0	8 (38) 2 (20)
**Tchekmedyian et al.** [206]	Vemurafenib + Anti-ErbB3	6	*BRAF-V600E (6)*	5	0	2 (40)
**NCT04554680** **—Japan**	Dabrafenib + Trametinib	5	*BRAF-V600E or RAS*	-	-	-
**NCT02152995** **—United States**	Trametinib	34	*BRAF or RAS*	-	-	-

a, all of the nine patients disclosed stable disease. Abbreviations: I, iodine; N, number; RAI, radioiodine.

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
