# Peer review of "Strategies for Radioiodine Treatment: What’s New"

_cancers, 2022, doi:10.3390/cancers14153800_

Round 1
Reviewer 1 Report
This review focuses on the advances in RAI therapy for DTC, involving mortality and recurrence risks, RAI treatment goals, pediatric DTC, side effects of RAI, and RAI refractoriness. The essentials of this redundant review keeps in line with 2015 ATA guidelines and 2022 ETA consensus and cites the latest publications, however own insights of the authors are scarce, which compromises the value of the paper to a certain degree. I would like to suggest the authors to refine the current draft and involve the following considerations:
1. Although the 2015 ATA guidelines extensively and intensively impact the academic attitude and clinical practice, please indicate the immaturity of the recurrence risk stratification in guiding RAI therapy, including the technical complexity, insufficiently proven rationality, and potential inability to explicitly assign appropriate candidates to each goal of RAI therapy, i.e. ablative, adjutant, and therapeutic. Then, point out that studies are needed to establish a more viable stratification system, involving serum thyroglobulin assessment, neck US, etc, to tailor RIA therapy for DTC.
2. The safety of RAI therapy remains a hot spot in RAI therapy with new references available, especially in pediatric and adolescent patients. Thus, explicit conclusions on the impact of RAI therapy on second malignancies are greatly needed, given scientifically analyses and objective determination.
3. Potential approaches to reduce side effects of RAI therapy, inflammation of the salivary gland and xerostomia in particular, are urgently needed, which is encouraged to be mentioned in the review.
4. By the way, it is of noting that 1.1 GBq of RAI brings comparable success rate of RRA to 3.7 GBq of RAI only in total thyroidectomized DTC patients with extremely limited remnant. When more remnant occurs, higher activities may be needed. Gradient dosing approach via quantifying thyroid remnant represents a feasible strategy (Thyroid. 2019 Jan;29(1):101-110.).
Author Response
This review focuses on the advances in RAI therapy for DTC, involving mortality and recurrence risks, RAI treatment goals, pediatric DTC, side effects of RAI, and RAI refractoriness. The essentials of this redundant review keeps in line with 2015 ATA guidelines and 2022 ETA consensus and cites the latest publications, however own insights of the authors are scarce, which compromises the value of the paper to a certain degree. I would like to suggest the authors to refine the current draft and involve the following considerations:
- Although the 2015 ATA guidelines extensively and intensively impact the academic attitude and clinical practice, please indicate the immaturity of the recurrence risk stratification in guiding RAI therapy, including the technical complexity, insufficiently proven rationality, and potential inability to explicitly assign appropriate candidates to each goal of RAI therapy, i.e. ablative, adjutant, and therapeutic. Then, point out that studies are needed to establish a more viable stratification system, involving serum thyroglobulin assessment, neck US, etc, to tailor RIA therapy for DTC.
Answer: Thank you for these comments. We agree that ATA guidelines are a perfectible tool. We added the above reflections in the final remarks, calling for future improvements. You can find these considerations in:
- Chapter 6, page 18: “Indication for RAI therapy mostly rely on ATA recurrence risk stratification which provides an estimate of the risk of persistent or recurrent disease after thyroid cancer surgery. This system is complex due to the numerous items to be taken into account and can be hard to apply in real life practice. To achieve a fine-tuned selection for ablative, adjuvant or therapeutic RAI and to overcome the present limitations of the ATA stratifications, an ongoing assessments with serum Tg and TgAb assessment and neck US may be considered. Furthermore, RAI protocols and timing are not standardized, resulting in a variety of behaviors according to single-centers practice. In this light, focused studies are needed to fill these gaps of evidence and improve our practice”
- The safety of RAI therapy remains a hot spot in RAI therapy with new references available, especially in pediatric and adolescent patients. Thus, explicit conclusions on the impact of RAI therapy on second malignancies are greatly needed, given scientifically analyses and objective determination.
Answer: Thanks for these remarks. It is true that growing concerns about oncological risks have been raised during the last years, notably in younger patients. The former manuscript version omitted these reflections due to the text length. However, we have now better specified these aspects, summarizing the latest available evidence, but also the existing controversies in this framework. We added as follows:
- Chapter 4, page 14: “Although extremely rare, the onset of RAI-related second malignancies represents a controversial and feared late AEs. Growing concerns have risen especially for younger DTC patients, due to the more aggressive therapeutic approaches applied and the long life expectancy. A study based on SEER large sample of 27050 pediatric DTC patients showed that the risk of second neoplasms appeared as concreate for both hematologic (RR=1.51; CI95: 1.08 to 2.01) - including leukemia (RR=1.92; CI95:1.04 to 3.56) - and solid cancers (RR=1.47; CI95:1.24 to 1.74) - including breast cancer (RR=1.46; CI95:1.10 to 1.95) [157]. Younger DTC survivors of ≥20 years disclosed the greater risks of secondary neoplasms, and RAI-related cancers were estimated to be 6% and 14% for solid and hematologic tumors, respectively [157]. Another SEER-based study showed a higher rate of second breast cancer in younger DTC patients who performed RAI therapy compared to both non-RAI treated (HR 1.65; CI95: 1.33–2.05, p < 0.001) and the general population (HR 1.21; CI95:1.02– 1.44, p < 0.05) [158]. However, breast cancer risk remains controversial and the above results have not been confirmed in a large and focused metanalysis of 200247 DTC patients treated or not with radioiodine [159]. Similarly, Kim et al [160] performed a propensity-score analysis on a wide retrospective multicenter sample of 24318 patients, including RAI-treated and non-RAI-treated subjects, concluding for the absence of significant risk in the occurrence of second tumors.”
- Chapter 6, page 18 : “On the other hand, due to the unclear risks of RAI-related second cancers, further insights are required and greater cautions should be exercised regarding the long-term safety of iodine.”
- Potential approaches to reduce side effects of RAI therapy, inflammation of the salivary gland and xerostomia in particular, are urgently needed, which is encouraged to be mentioned in the review.
Answer: Thanks for your comment. As stated above, due to the text length we limited some reflections. Under reviewers’ request, we have now added these insights and also suggested further research in this field.
- Chapter 4, pages 14-15: “Considering the high rate of salivary gland AEs, several preventive strategies have been investigated in pre-clinical and clinical settings [131,161]. Pre-clinical studies explored the efficacy of various agents, including antioxidants (i.e. Vitamin E) and nutraceutics (i.e. Curcuma longa, Ocimum sanctum, and zinc), showing potential benefits in non-human subject tests [161]. A clinical trial analyzing the influence of Vitamin C on salivary glands’ iodine absorption found only limited effects, irrespective of the moment of administration during RAI therapy [162]. Similarly, pilocarpine, a parasympathomimetic drug stimulating salivary glands, didn’t prove substantial advantages, with potential severe toxicities in selected categories, i.e. patients with asthma or cardiac diseases [161]. Amifostine, a supposed salivary glands radioprotector, showed potential effectiveness in two clinical trials [163,164], but failed to confirm the same results in a systematic review [116], and its tolerance and costs further limited its applications in clinical practice [161]. Potential benefits of sialagogues agents, such as lemon candies and lemon juices, have been reported thanks to the increase in salivary function and, thus, in iodine washout [161]. However, some concerns emerged after the evidence of a higher rate of AEs in early lemon candies eaters (<24h after iodine administration), due to the concurrent risen of blood flow to the glands, which potential increases iodine uptake and retention [165]. This event, also named the “rebound effect”, hasn’t been definitely proven [166,167], but the timing of sialagogues administration remains controversial, as well as potential rebounds during the first 24 hours from iodine delivery [161,166,168].”
- Chapter 6, page 18: “Some safety issues still wait for clarifications. Prevention strategies are urgently needed for frequent RAI AEs, such as xerostomia and salivary gland impairment.”
- By the way, it is of noting that 1.1 GBq of RAI brings comparable success rate of RRA to 3.7 GBq of RAI only in total thyroidectomized DTC patients with extremely limited remnant. When more remnant occurs, higher activities may be needed. Gradient dosing approach via quantifying thyroid remnant represents a feasible strategy (Thyroid. 2019 Jan;29(1):101-110.).
Answer: Thank you for this remark. As suggested, we added these considerations in:
- Paragraph 2.1, pages 4-5: “However, patients with very large thyroid remnants might require higher iodine activities to achieve a complete remnant ablation. At this regard, Jin et al. [31] showed that an approach in which the activity of RAI therapy is based on Tg values and on RAI uptake on a diagnostic RAI scan can achieve a better ablation rate compared with a fixed activity. This approach might be appropriate for those subjects with huge residual tissue eventually avoiding the need of repeated RAI treatments.”
Reviewer 2 Report
The manuscript by Sparano and colleagues summarizes very well the current status of radioiodine therapy in DTC and allows an overview of the current trial situation. The manuscript is well organized and written. Nevertheless, I have some points that I ask the authors to revise.
1. please use concentration and not dosage for Tg (pages 5 and 8)
2. the acronyms/abbreviations to table 1 are incorrect or not used in the table: PTMC and ETE are not used at all in the table, but can be found among the acronyms. RRA is used as a heading in column 10, but is not given as an acronym, but instead RAIT, which I would also rather use as a heading here. In addition, there are spelling errors in the table (i.e., extrathyroidal) that should be corrected. Why does the column on tumor stage (T) have (size) in parentheses in some places? This is not understandable and would need to be explained.
Please optimize the format in most tables. Some words are cut off or not correctly separated. In table 2, the date of publication could also be removed from the authors to generate more space.
4. I do not agree with the statement that radioiodine therapy is the treatment of choice for distant metastases. In my view, this depends on how extensive the distant metastasis is, where the distant metastasis is located, the patient's health status, and whether a distant metastasis acts as a pacemaker for the disease. Solitary metastases may well be amenable to a surgical approach, especially brain metastases, so these patients should be discussed in an interdisciplinary endocrine tumor board, after which the therapeutic options (radioiodine therapy versus surgery) must be discussed with the patient.
5. please correct bones in bone on page 9
6. page 13: Please write out the abbreviation for NIS the first time it is used.
7. page 15: please write out the abbreviation for HCC the first time it is used. The reader cannot directly recognize that this stands for Hürthle Cell Carcinoma.
8. please always use a hyphen for BRAF-V600E (on page 15 it is written without hyphen)
9. page 16: better use downstream instead of following in the sentence "activating the following kinases MEK and ERK".
Author Response
The manuscript by Sparano and colleagues summarizes very well the current status of radioiodine therapy in DTC and allows an overview of the current trial situation. The manuscript is well organized and written. Nevertheless, I have some points that I ask the authors to revise.
- please use concentration and not dosage for Tg (pages 5 and 8)
Answer: Thank you for this remark. As suggested, the term dosage has been substituted by concentration at pages 4 and 8.
- the acronyms/abbreviations to table 1 are incorrect or not used in the table: PTMC and ETE are not used at all in the table, but can be found among the acronyms. RRA is used as a heading in column 10, but is not given as an acronym, but instead RAIT, which I would also rather use as a heading here. In addition, there are spelling errors in the table (i.e., extrathyroidal) that should be corrected. Why does the column on tumor stage (T) have (size) in parentheses in some places? This is not understandable and would need to be explained.
Please optimize the format in most tables. Some words are cut off or not correctly separated. In table 2, the date of publication could also be removed from the authors to generate more space.
Answer: We apologize for these mistakes. Table 1 has now been amended and all the acronyms have been adequately introduced. All the tables have been optimized.
- I do not agree with the statement that radioiodine therapy is the treatment of choice for distant metastases. In my view, this depends on how extensive the distant metastasis is, where the distant metastasis is located, the patient's health status, and whether a distant metastasis acts as a pacemaker for the disease. Solitary metastases may well be amenable to a surgical approach, especially brain metastases, so these patients should be discussed in an interdisciplinary endocrine tumor board, after which the therapeutic options (radioiodine therapy versus surgery) must be discussed with the patient.
Answer: Thank you for these comments. We agree that some conditions (i.e. unique metastasis, bulky or threatening) require specific considerations, notably within a multidisciplinary tumor board. For these reasons, we mitigate the former statement, as follows:
- Paragraph 2.3, page 9: “RAI therapy is the favored starting approach for metastatic DTC [2] . All metastatic DTC cases should be discussed in a multidisciplinary tumor board. The presence of bulky or threatening metastatic sites should lead to consider focal treatments (e.g. surgery, interventional radiology, radiotherapy) as an alternative or in association to RAI therapy, as well as in case of single or oligo metastatic disease in order to limit repeated RAI administrations.”
- please correct bones in bone on page 9
Answer: Thank you for this remark. We corrected the word within the text.
- page 13: Please write out the abbreviation for NIS the first time it is used.
Answer: We apologize for this mistake. The acronym has now been adequately introduced at page 13.
- page 15: please write out the abbreviation for HCC the first time it is used. The reader cannot directly recognize that this stands for Hürthle Cell Carcinoma.
Answer: We apologize for this mistake. The acronym has now been adequately introduced now at page 16.
- please always use a hyphen for BRAF-V600E (on page 15 it is written without hyphen)
Answer: Thank you for this remark. The word has been corrected.
- page 16: better use downstream instead of following in the sentence "activating the following kinases MEK and ERK".
Answer: Thank you for this remark. The term has been substituted as suggested.
Reviewer 3 Report
In the present review, Clotilde Sparano and co-workers provides insights into the most recent and high-quality evidence on radioactive iodine treatment (RAI). Indeed, the authors concluded that, despite the achievements which had been made, several issues still need to be addressed in terms of RAI indications and protocols, heading toward the RAI strategy of the future.
Overall, I think that the paper is well-written, timely, and it could be of interest to the readers of "Cancers" and researchers, in general. I would like to congratulate the authors on their work.
I have only a little question/curiosity: in light of the recent evidence, please to discuss on the possible application of nutraceutics and/or antioxidants/antinflammatory compounds to prevent complications and side effects of RAI.
Author Response
In the present review, Clotilde Sparano and co-workers provides insights into the most recent and high-quality evidence on radioactive iodine treatment (RAI). Indeed, the authors concluded that, despite the achievements which had been made, several issues still need to be addressed in terms of RAI indications and protocols, heading toward the RAI strategy of the future.
Overall, I think that the paper is well-written, timely, and it could be of interest to the readers of "Cancers" and researchers, in general. I would like to congratulate the authors on their work.
I have only a little question/curiosity: in light of the recent evidence, please to discuss on the possible application of nutraceutics and/or antioxidants/antinflammatory compounds to prevent complications and side effects of RAI.
Answer: Thank you for these comments. As already requested, we have now discussed the use of these compounds to prevent iodine adverse events.
- 4, pages 14-15: “Considering the high rate of salivary gland AEs, several preventive strategies have been investigated in pre-clinical and clinical settings [131,161]. Pre-clinical studies explored the efficacy of various agents, including antioxidants (i.e. Vitamin E) and nutraceutics (i.e. Curcuma longa, Ocimum sanctum, and zinc), showing potential benefits in non-human subject tests [161]. A clinical trial analyzing the influence of Vitamin C on salivary glands’ iodine absorption found only limited effects, irrespective of the moment of administration during RAI therapy [162]. Similarly, pilocarpine, a parasympathomimetic drug stimulating salivary glands, didn’t prove substantial advantages, with potential severe toxicities in selected categories, i.e. patients with asthma or cardiac diseases [161]. Amifostine, a supposed salivary glands radioprotector, showed potential effectiveness in two clinical trials [163,164], but failed to confirm the same results in a systematic review [116], and its tolerance and costs further limited its applications in clinical practice [161]. Potential benefits of sialagogues agents, such as lemon candies and lemon juices, have been reported thanks to the increase in salivary function and, thus, in iodine washout [161]. However, some concerns emerged after the evidence of a higher rate of AEs in early lemon candies eaters (<24h after iodine administration), due to the concurrent risen of blood flow to the glands, which potential increases iodine uptake and retention [165]. This event, also named the “rebound effect”, hasn’t been definitely proven [166,167], but the timing of sialagogues administration remains controversial, as well as potential rebounds during the first 24 hours from iodine delivery [161,166,168].”
- 6, page 18: “Some safety issues still wait for clarifications. Prevention strategies are urgently needed for frequent RAI AEs, such as xerostomia and salivary gland impairment.”
Round 2
Reviewer 2 Report
The authors have adequately commented on all my comments and revised the manuscript.